

# Spatiotemporal assessment of landslide risk over large areas: A case study of the Valencian Community (1950–2021)

Isidro Cantarino Marti[1], Miguel Ángel Carrión Carmona[1], Eric Gielen[2], José-Sergio Palencia-Jiménez [2]

[1] Department of Geological and Geotechnical Engineering, Universitat Politècnica de València, Valencia, 46022, Spain
[2] Department of Urbanism, Universitat Politècnica de València, Valencia, 46022, Spain

*Correspondence to:* Miguel Ángel Carrión Carmona (mcarrio@upv.es)

**Abstract.** The risk posed by natural hazards has gained growing attention in recent decades, largely due to the intensification and recurrence of extreme events, with the climate crisis identified as the primary driver. Landslide risk is no exception, although its impacts are generally less evident than those of floods or, particularly, severe droughts. In both cases, urban expansion has further exacerbated the problem, especially since the mid-twentieth century in more developed regions. This residential growth often took place in poorly regulated settings, particularly during its early stages, leading to the occupation of areas that were environmentally, culturally, or from a landscape perspective unsuitable, and frequently exposed to natural hazards. In fact, the risk of landslides affecting buildings located on susceptible terrain can largely be attributed to ineffective land management, often resulting from the absence of specific regulations. This study introduces a set of risk indices that serve as objective tools for the dynamic assessment of landslide risk in extensive and spatially fragmented territories divided into local entities. Based on these indices, criteria are proposed to evaluate the degree of risk and the adequacy of its management within each local entity, considering the evolution of urban development. Finally, a classification system is presented that organizes all cases according to their severity, offering a decision-support tool for public authorities tasked with ensuring effective land management.

## 1 Introduction

Urban expansion is a phenomenon intrinsically linked to the development of human societies, particularly since the Industrial Revolution, and is inherent to the growth of cities. Today, this process is also associated with improvements in living standards, transportation, communications infrastructure, and services beyond traditional population centers. Unsurprisingly, urban development has been analyzed from multiple perspectives, with urban planning recognized as a fundamental tool to regulate and organize such growth. Indeed, in the seminal work of Fernando de Terán (1982), urban planning was identified as essential to bring order on urban growth, in light of the damage and disruptions caused by unregulated development.

It is well established that poorly planned urban expansion increases exposure to natural hazards. Along with climate change, one of the main factors explaining the rising risks in residential areas is the urban expansion with global evidence indicating an increasing incidence of hazard, particularly landslides (Chen et al., 2024; Haque et al., 2019; Zhou and Zhao, 2013). These processes have significantly increased pressure on land and, consequently, on populations, due to the occupation of terrain unsuitable for residential construction (Fernández Arce et al., 2018). This situation reflects inadequate land management, largely resulting from the absence of adequate hazard zoning policies that would enable proper land-use planning (Cascini et al., 2005).

The growing exposure of urban areas to natural hazards is therefore linked to the lack of integration of hazard considerations into urban planning. Structural solutions cannot be considered the primary strategy for risk reduction and must be complemented with passive measures (Corominas, 2013). Proper planning can reduce exposure within urban areas, which is especially relevant in developing countries (Caleca et al., 2024). Urban planning is also considered a powerful tool to achieve efficient and equitable adaptation between land occupation and natural hazard risk (Hamma and Petrişor, 2018; Macintosh, 2013).

Landslide disasters, in particular, can have severe consequences, including loss of life, damage to buildings and infrastructure, and environmental impacts. Effective management of landslide risk requires the involvement of multiple stakeholders and the adoption of an integrated disaster risk management approach. This involves a complex process aimed at predicting, reducing, and permanently controlling the factors that trigger such hazards, while simultaneously pursuing sustainable human, economic, and environmental development (Alcántara-Ayala and Sassa, 2023).



Landslides are among the most hazardous natural disasters worldwide, both in frequency and severity, causing widespread loss of life and damage to infrastructure. Their incidence has increased notably in recent decades (Cascini et al., 2005; Lee et al., 2017; Sandić et al., 2017). Numerous studies highlight the interactions between landslides and urban development (Johnston et al., 2021). In some cases, land-use regulations exist but are not enforced, leading to illegal or irregular settlement, as in the Campania region of Italy (Di Martire et al., 2012). In other cases, the expansion of urban areas has altered river courses, thereby exacerbating landslide hazards, as documented in Genoa, Italy (Faccini et al., 2015) and Doboj, Bosnia and Herzegovina (Sandić et al., 2017).

In many instances, the development of tourism facilities and secondary residences, which demand large amounts of land, has driven settlement in areas unsuitable due to environmental, cultural, or hazard-related factors. Weak or absent administrative controls have allowed development in previously overlooked areas, leading to landslides. For example, Katsigianni and Pavlos-Marinos (2017) reported such a case on the Greek island of Santorini. A similar situation occurred in Mengshan, China (Peng and Wang, 2015), where engineering measures were implemented only after the construction of high-risk tourist developments. In the Caribbean, informal urban expansion and associated deforestation have been identified as major landslide triggers (Bozzolan et al., 2023). In this context, urban planning must take these factors into account to guide land development and avoid uncontrolled expansion.

Given this scenario, which is prevalent worldwide, risk governance integrated into urban planning is urgently required (Renn and Klinke, 2013) to enhance the resilience of urbanized areas and their future growth (Zhai et al., 2015). Proper hazard zoning is also required to support disaster risk reduction (Wang et al., 2008). Urban governance faces the major challenge of developing effective systems and tools suited to evolving natural hazard contexts (Birkmann et al., 2014). Experiences in some countries have demonstrated the importance of engaging communities in this issue, promoting the adoption and implementation of solutions, as observed in New Zealand (Gough, 2000).

This situation is particularly relevant in mountainous coastal zones, especially along Mediterranean shorelines. Di Martire et al. (2012) note the problem in Italy, while García et al. (2003) emphasize that new coastal construction often disregards planning regulations, which have proven ineffective in controlling this phenomenon. For this reason, strict enforcement of regulations and the implementation of effective control mechanisms are essential.

This issue has been extensively addressed in the scientific literature. Landslide risk evaluation, management, and mitigation have been extensively studied over the past decades. Several notable works include the synthesis by Dai et al. (2002), which provided a critical review of landslide research and loss-reduction strategies, as well as the relevant contributions of Lee and Jones (2004) and Glade and Crozier (2005), who offered a multidisciplinary perspective on landslide management. Another important reference is the comprehensive review of quantitative risk assessment methods by Corominas et al. (2014).

Despite this extensive body of research, it is noteworthy that urban planning and regulatory measures in landslide-prone areas have received less attention compared to structural solutions (Corominas, 2013). Ultimately, it is not only necessary to quantify and map risks and to propose avoidance or mitigation measures, but also to establish procedures to monitor the effective implementation of these measures and to evaluate whether risk reduction is actually achieved. The key question is whether the rate of settlement in hazard-prone zones eventually stabilizes or declines. If so, this turning point must be defined by the strict enforcement of specific regulations that prohibit or restrict residential development in such areas.

In line with these considerations, the present study aims to establish objective criteria to evaluate the status and evolution of landslide risk across large territories, applied to a study area of more than 22,000 km², using indicators that are simple to derive and interpret. To ensure comparability, a consistent calculation procedure was applied across the entire territory. Particular attention was devoted to assessing the adequacy of urban development processes within local entities in relation to landslide hazard exposure. Finally, the study proposes monitoring tools for public authorities tasked with ensuring compliance.

## 2 Methodology

The analysis of the status and evolution of landslide risk affecting residential buildings across large areas requires the use of comparable indicators, which must also be calculated over defined time periods to capture temporal changes. This study proposes a two-phase approach. First, the risk is assessed. Risk calculation requires evaluating the interaction between hazard (probability of occurrence), exposure (value of the elements at risk), and vulnerability (severity of potential damage). This interaction is expressed through the well-known risk equation, generally quantified in economic units:

$$Risk\ Value\ (RV) = Hazard\ (H) \times Exposure\ (E) \times Vulnerability\ (V) \tag{1}$$





Second, dimensionless risk indices are obtained based on surface area and risk values. These indices constitute the main contribution of this study, as they are designed to serve as control tools for landslide risk management. The procedure requires cartographic information dividing the territory into areas of landslide susceptibility. These are the widely used Landslide Susceptibility Maps (LSM), which delineate such areas through a well-established methodology (Corominas et al., 2014). This mapping commonly classifies the territory into five

susceptibility levels (Landslide Susceptibility Index, LSI), ranging from very low (LSI level 1 or L1) to very high (LSI level 5 or L5). A "risk zone" is defined as the area classified with medium-to-high susceptibility, corresponding to levels L3, L4, or L5. Levels L1 and L2 are generally not affected by landslides, although this must be verified against available inventories. In addition to susceptibility mapping, a landslide inventory is required to capture their spatiotemporal distribution. Finally, an economic valuation of the affected residential

buildings is needed. The complete process is illustrated in the flow chart presented in Fig. 1, which is further explained throughout Section 2.

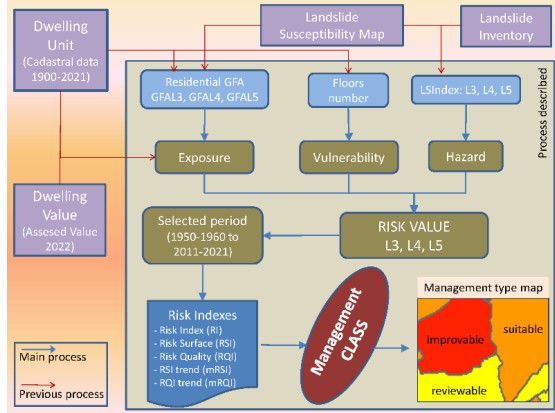

**Figure 1.** Flow chart of the methodology (see explanation in Section 2).

### 2.1 Landslide risk calculation

As indicated above, the calculation of risk requires evaluating the interaction among the factors included in Eq. (1). The unit of analysis is defined as the *Local Entity (LE)*, into which the study area is divided, each with its own administrative capacity and decision-making authority. This corresponds to the *Local Administrative Unit (LAU)* defined by the European Union as the smallest administrative division, which in Spain corresponds to municipalities.

For the minimum unit of calculation, geolocated polygons of residential buildings are required, at the level of the *Dwelling Unit (DU)*, which provide the constructed habitable area or *Ground Floor Area (GFA)*. In Spain, these units are defined as *Cadastral Parcels*.

Accordingly, the economic value of risk is obtained from Eq. (1), extended to the scale of a Local Entity (or municipality, in Spanish notation). It is calculated based on the dwelling units ($DU_i$) it contains, as expressed

in the following equation:

$$RV_{LE} = \sum RV_{DUi} = \sum H_{DUi} \times E_{DUi} \times V_{DUi} \tag{2}$$

### 2.1.1 Hazard

Hazard is considered as the probability of a landslide occurring within a specific location and time period;

therefore, both components must be accurately determined. It is directly related to the landslide susceptibility index (LSI) defined in the landslide susceptibility maps (LSM) (Lee, 2009). In addition, a landslide inventory is required to calculate both temporal (annual) and spatial probabilities.

$$H_{DU} = f(LSI) = Ps \times Pa \tag{3}$$

The spatial probability ($P_s$) is calculated for a given medium-to-high susceptibility level *i* as:




$$Ps\,(i) = \frac{SRLi}{SLi} \times Faj \,, \quad for\ i = L3, L4, L5$$

(4)

where $SLi$ is the total surface area of level $i$, and $SRLi$ is the surface area of level $i$ affected by landslides. The adjustment factor ($Faj$) accounts for whether the event occurs in an inhabited area, derived from the ratio between built surface within risk zones and total risk surface.

The annual probability ($P_a$) is also calculated for each level $i$ as:


$$Pa(i) = \left( \frac{number\ of\ record\ events}{number\ of\ years\ of\ record} \right) \times Gaj$$

(5)

This annual probability ($P_a$) must be adjusted using the factor $Gaj$, which reflects the magnitude of the inventoried landslides. The use of $Gaj$ must be carefully considered, since inventories are often incomplete: many landslides are not officially recorded if they occur away from populated areas. In other words, inventories already underestimate the true number of events. Consequently, the estimation of landslide probability must ultimately

rely on expert judgment, making use of the available data, knowledge, and experience (Lee, 2009).

**2.1.2 Exposure**

Exposure requires the valuation of the elements at risk in monetary units. In this study, the affected elements are defined as residential buildings or dwelling units located within the study area. The valuation is generally based,

first, on the constructed and habitable surface area of each dwelling unit (*Ground Floor Area, GFA*), since the land value itself is not considered to be directly affected. In addition, the reconstruction cost of the dwelling, or *Dwelling Unit Value (DV)*, expressed in economic units per unit of surface area, must also be considered. Accordingly, the reconstruction value used to obtain the exposure of each dwelling unit within a risk zone is determined by the following equation:


$$E_{DU} = GFA_{DU} \times DV$$

(6)

**2.1.3 Vulnerability**

Following a technical and engineering-based approach, physical vulnerability is defined as the severity of damage sustained by the exposed elements. Vulnerability is a function of the magnitude or intensity of the landslide (*Landslide Magnitude, LM*) and depends on the resistance capacity of the affected element, which is closely related

to building height. To determine $LM$ within a territory, it is essential to compile a landslide inventory that identifies the main types of landslides, their morphometric parameters, velocity, and the associated observed damages. The other factor required to evaluate vulnerability is the estimation of the resistance of residential buildings (*Building Resistance, BR*), considering construction type, materials, age, and height (Kappes et al., 2012; Pereira et al., 2020; Singh et al., 2019). The formula applied to each dwelling unit is:


$$V_{DU} = LM \times \left( 1 - BR_{DU} \right)$$

(7)

According to Papathoma-Köhle et al. (2017), in their study on debris flows, expected losses decrease as building height increases, which justifies assigning higher $BR$ values in such cases.

**2.2 Risk indices and qualifiers**

Previous applications of indices at "very large areas" (national or regional scale), but estimating risk at the local entity level, can be found in two notable studies. First, Pereira et al. (2020) evaluated the entire territory of Portugal, defining a *Landslide Risk Index (LRI)* that incorporates population density together with the total number of buildings per municipality. Second, Segoni and Caleca (2021) examined the Italian peninsula, using *soil sealing* as a variable to represent the extent of built-up areas. However, given the broad territorial scale, neither study

incorporated the specific location of population or buildings with detailed economic valuation, nor did they account for temporal intervals of risk. In the present study, all residential buildings at the *Dwelling Unit (DU)* level were considered, together with their reconstruction value ($DV$) and susceptibility level ($L_n$), in order to achieve greater precision in quantitative risk assessment and to evaluate temporal variation trends.



### 2.2.1 Quantitative risk indices

Within this category, numerical indices provide information on the current state of risk as well as its evolution over time. Geolocated residential construction data and the aforementioned Landslide Susceptibility Maps (LSM) are required.

*State indices*

These indicators provide a fixed value for a specific moment in time, which can be updated for successive periods in order to analyze trends and particular situations. In a previous study (Cantarino et al., 2021), an indicator was developed based on the annual ratio between the assumed risk value ($RV$) and the constructed surface area ($GFA$) within a given local entity ($LE$). This indicator was referred to as the *Risk Ratio (RR)*. Accordingly, when $\Delta RR > 1$ (increasing function), construction in risk zones is increasing; whereas when $\Delta RR < 1$ (decreasing function), the growth of risk is lower than that of construction, meaning that new developments are avoiding risk zones, which is the desirable outcome. Although efficient for trend analysis, this indicator is not dimensionless and its practical meaning is not straightforward. For this reason, a dimensionless and more intuitive index is introduced: the *Risk Index (RI)*, which can be considered an evolution of RR. This index is defined as the ratio between the calculated risk value ($RV$, in monetary units) for each LE and the theoretical maximum risk value ($RV_H$), assuming that all residential buildings were located in susceptibility level 4 (high, L4 or H) with average vulnerability ($Vm$) according to the LSM. It can be interpreted as an approximation of the "percentage of maximum possible risk" for an LE, yielding values typically within decimals. Level L5 was not considered, as it is less realistic due to its smaller extent and would result in excessively high $RV_H$ values and, consequently, very low $RI$ values. For a given LE, the $RI$ value at a specific time is calculated as:

$$RI = \frac{RV}{RV_H} \tag{8}$$

That is, the ratio between the calculated risk value for the considered area or period, and the theoretical risk value if all residential buildings of the LE were located in the high susceptibility zone (L4), combined with the economic value of all dwellings ($E$) and average vulnerability ($Vm$). A high $RI$ value therefore indicates that the majority of residential surface is located in susceptibility level L4 ($GFA_R$L4). Expanding Eq. (2) for susceptibility levels $i$ (medium-to-high):

$$RV = \sum Hi \times Vi \times Ei = \sum Hi \times Vi \times GFA_R(i) \times DVi, \qquad i = L3...L5 \tag{9}$$

$$RV_H = H_H \times Vm \times E = H_H \times Vm \times GFA \times DV \tag{10}$$

where $RV$ is obtained through the hazard, exposure, and vulnerability of each affected dwelling unit, and exposure is defined as the constructed residential area in each risk zone ($GFA_R$) multiplied by the reconstruction cost per unit of surface ($DV$) (see Eq. (6)). An important feature of this index is the possibility of deriving two highly useful partial indices. By multiplying and dividing the previous expression by the constructed surface in risk zones ($GFA_R$), we obtain:

$$RI = \frac{GFA_R \times RV}{GFA_R \times \left(H_H \times V_m \times GFA \times DV\right)} \tag{11}$$

Rearranging yields:

$$RI = \frac{GFA_R}{GFA} \times \frac{RV}{H_H \times V_m \times GFA_R \times DV} \tag{12}$$

Thus, $RI$ can be expressed as the product of two components: the *Risk Surface Index (RSI)* and the *Risk Quality Index (RQI)*:

$$RI = RSI \times RQI \tag{13}$$



$$RSI = \frac{GFA_R}{GFA} \qquad (14)$$


$$RQI = \frac{RV}{H_H \times Vm \times GFA_R \times DV} \qquad (15)$$

The *RSI* reflects the proportion of built surface under risk relative to the total constructed area. It can approach unity in small LEs with little residential surface almost entirely exposed to landslide susceptibility. The *RQI*, on the other hand, indicates whether the risk value approaches its theoretical maximum. Eq. (15) can be further developed assuming that residential building typologies are similar within a LE, such that vulnerability is
constant ($V_i = Vm$) and *DV* is uniform. Considering the hazard probability ratios (*pR*) between susceptibility levels defined in the LSM:

$$pR4 = \frac{H(L3)}{H(L5)} \qquad (16)$$

$$pR5 = \frac{H(L4)}{H(L5)} \qquad (17)$$

Then, for levels L3, L4, and L5:

$$RQI = \frac{H(L3) \times GFA_R L3 + H(L4) \times GFA_R L4 + H(L5) \times GFA_R L5}{GFA_R H(L4)} = \qquad (18)$$

$$= \frac{pR4 \times GFA_R L3 + GFA_R L4 + \frac{1}{pR5} \times GFA_R L5}{GFA_R} =$$

$$= \frac{pR4 \times pR5 \times GFA_R L3 + pR5 \times GFA_R L4 + GFA_R L5}{GFA_R \times pR5}$$

This can be simplified when no built surface exists in L5:

$$RQI = = \frac{pR4 \times GFA_R L3 + GFA_R L4}{GFA_R} \qquad (19)$$

This simplification of *RQI* clearly shows that its value depends mainly on the built surface located in
high-susceptibility zones (*GFA_RL4*), since *pR4* is less than one. Moreover, the *RQI* value provides insight into whether construction is concentrated in high-susceptibility zones (level L4) and its evolution. If the total *GFA* increases in the same proportion as the surface at level L4, the *RQI* value remains constant.

*Evolution indices*

The indices described above do not provide information on the temporal dynamics of risk. To capture construction
trends in risk zones, linear regression slopes were calculated from the latest values in the available time series. This provides insight into whether risk in a given area is stable, increasing, or decreasing. The slopes in this study are expressed in sexagesimal degrees. Since two series of state values (*RSI* and *RQI*) are available, it is more informative to compute slopes separately for each, rather than only for *RI*. Thus, two evolution indices are defined: *mRSI* and *mRQI*. The meanings differ: *mRSI* is positive when growth in total risk surface (*GFA_R*) exceeds that of
total built surface (*GFA*); if lower, it becomes negative. This index does not account for susceptibility level. Conversely, *mRQI* is positive when the proportion of *GFA_RL4* (and *GFA_RL5*) increases relative to *GFA_RL3*, indicating higher landslide probability and therefore higher risk values.

*Interpretation of indices*

After defining both state and evolution indices, it is useful to summarize their meaning. Table 1 presents the indices
used, their ranges of variation, and general considerations. In general, it can be stated that the highest values of these indices are found in local entities with limited built surface located in mountainous areas. Indeed, because these municipalities provide lower absolute values, it is easier to reach higher percentages and ratios. To illustrate





the order of magnitude, two scenarios are proposed for two local entities (LE1 and LE2), both with the same initial *GFA* (10 units of surface) but evolving differently between *t1* and *t2*. They differ in the proportion of surface in
risk zones (*GFA_R*) versus outside risk (*GFAo_R*). Using Eq.s (14) and (18), and assuming *pR4 = 0.1*, the resulting indices are summarized. The complete analysis of Fig. 2 is shown in Table 2, applicable to cases where two LEs share similar *GFA* values but differ in landslide risk exposure. According to Table 2, *RSI* provides an indication of the magnitude of the problem in an LE but does not fully reflect actual risk. Meanwhile, *mRSI* does not provide a definitive conclusion on risk value growth and should be regarded as a secondary or complementary index.

**Table 1.** Risk indices: ranges of variation and general considerations.

| Index | Range | Considerations |
|---|---|---|
| RSI | [0, 1] | Indicates the proportion of built surface in risk zones relative to the total. Levels L3, L4, and L5 are not distinguished. |
| RQI | [pR4, 1/pR5] [1] | Indicates the proportion of built surface in levels L4 and L5 relative to total surface in risk zones. High values occur in LEs where most construction is concentrated in L4 and L5. |
| mRSI | [-59°, +59°] | Indicates the evolution of RSI. High positive values indicate major increases in built surface within risk zones. However, if growth occurs mainly in L3, this will not substantially increase risk. |
| mRQI | [-56°, +56°] | Indicates the evolution of RQI. High positive values occur when construction in L4 and L5 increases with little growth in L3, clearly implying increased risk. |

[1] A unit value occurs when all built surface is located in level L4. It may exceed one if construction also occurs in level L5.

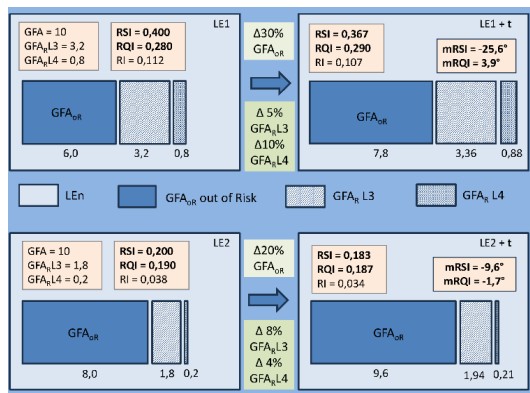

**Figure 2.** Indices for different scenarios of landslide risk status and growth.

**Table 2.** Comparison of indices between LE1 and LE2 (based on Fig. 2).

| Comparison | Interpretation |
|---|---|
| RSI$_1$ > RSI$_2$ | LE1 has a higher proportion of built surface in risk zones relative to total construction. LE1 will only have higher risk if its surface in L4 is greater. |
| RQI$_1$ > RQI$_2$ | LE1 has a higher proportion of construction in L4. LE1 therefore has higher risk. |
| mRSI$_2$ > mRSI$_1$ | LE2 increases its proportion of surface in risk zones more during *t1–t2*. However, LE2 will only have higher risk growth if more surface is built in L4, which is not the case in Fig. 2. |
| mRQI$_1$ > mRQI$_2$ | LE1 increases its proportion of construction in L4 more during *t1–t2*. LE1 therefore increases its risk more than LE2. |

**2.2.2 Qualitative risk qualifiers**

All indices must be organized in a consistent manner to enable direct comparison between local entities (LEs). The objective is to identify the most relevant cases across large territories subdivided into numerous LEs. In this way, priority areas for intervention and monitoring can be selected. According to the interpretations discussed in Section 2.1.3 and Table 2, the preferred indices are those based on risk quality and, particularly, on its evolution
(*RQI* and *mRQI*). The *RSI* index is also relevant, as it reflects the overall extent of built-up surface within risk zones. Table 3 lists all indices, with shading used to highlight those of greater interest.





**Table 3.** Quantitative risk indices (preferred indices highlighted).

| Category | Global risk | Risk quantity | Risk quality |
|---|---|---|---|
| State | RI | RSI | RQI |
| Trend | mRI | mRSI | mRQI |






For a large number of LEs, frequency-based levels can be defined to highlight the highest values within a given territory. Thus, the quantitative values provided by the preferred indices in Table 3 must be converted into qualitative qualifiers to establish a specific *Management Code (MC)*. This code should concisely convey the quality of results. Five classes are proposed: Very High (VH), High (H), Medium (M), Low (L), and Very Low (VL). Class boundaries are defined by percentiles P90, P60, P40, and P20. These restrictive thresholds are designed to minimize type I errors (false positives). Percentiles may be calculated for the entire study area or for sufficiently large subareas, to account for local singularities. LEs with low *RI* values must be excluded to avoid biasing the quantile results downward. This classification is structured into five levels (A, B, C, D, E) for *RQI* and *mRQI*. Due to its lower relevance, the *RSI* index (see Table 3) is simplified into two levels only: high–very high ("a") and medium–very low ("b"). The threshold distinguishing both levels is set at percentile P60. Based on these levels, five management types are defined: Very improvable (VI), Improvable (I), Reviewable (R), Suitable (S), and Unaffected (U, for entities with insignificant risk values). In addition, a low proportion of surface under risk (*RSI = b*) reduces the management type classification by one level. Indeed, a LE with little construction in risk zones is less likely to present significant problems, and in some cases, this could reflect methodological error. Consequently, the main management types (qualifiers) are summarized in Table 4. Codes in parentheses indicate cases downgraded one level due to low *RSI*. From these main codes, all combinations of *mRQI* and *RQI* levels can be derived, with descending importance assigned as *mRQI → RQI → RSI*, in line with the criteria outlined above. The final distribution is shown in Table 5, with shading highlighting the two combinations requiring the most attention. According to this criterion, combinations including an *RSI = b* are downgraded by one level. For instance, an entity classified as AA with *RSI = a* falls into "Very improvable (VI)"; however, if it has *RSI = b*, it is downgraded to "Improvable (I)". Similarly, entities classified as "Suitable (S)" are reclassified as "Unaffected (U)" if *RSI = b*.

**Table 4.** Main classification types.

| Class | Level | | | Code management | Management type |
|---|---|---|---|---|---|
| | mRQI | RQI | RSI | | |
| Very High | A | A | a | AAa | Very improvable (VI) |
| High | B | B | a, (b) | BBa, (AAb) | Improvable (I) |
| Medium | C | C | a, (b) | CCa, (BBb) | Reviewable (R) |
| Low | D | D | a, (b) | DDa, (CCb) | Suitable (S) |
| Very Low | E | E | a, b | EEa, EEb, (DDb) | Unaffected (U) |

**Table 5.** Management type classification.

| RQI levels ↓ / mRQI levels → | A | B | C | D | E |
|---|---|---|---|---|---|
| A | Very improvable (VI) | Improvable (I) | Reviewable (R) | Suitable (S) | Unaffected (U) |
| B | Very improvable (VI) | Improvable (I) | Reviewable (R) | Suitable (S) | Unaffected (U) |
| C | Improvable (I) | Improvable (I) | Reviewable (R) | Suitable (S) | Unaffected (U) |
| D | Reviewable (R) | Reviewable (R) | Reviewable (R) | Suitable (S) | Unaffected (U) |
| E | Suitable (S) | Suitable (S) | Suitable (S) | Suitable (S) | Unaffected (U) |

**3 Application to the Valencian Community (Case Study)**

**3.1 Study area**



The Valencian Community is an autonomous region of Spain located in the eastern and southeastern Iberian Peninsula, along the Mediterranean coast. Covering 23,255 km², it is the eighth largest region in Spain by surface area and represents 4.6% of the national territory. The inland areas are mountainous, with peaks exceeding 1,800 m in elevation. Its complex orography is shaped by the proximity of the sea, with a fluvial system that carves into mountain headwaters and expands into alluvial plains toward the coast. This narrow, elongated territory extends in a north–south direction, bordered by Tarragona to the north and Murcia to the south, and bounded to the east by the Mediterranean Sea. It includes the provinces of Castellón (CST), Valencia (VLC), and Alicante (ALC), which



together comprise a total of 542 local entities (municipalities). According to 2023 data from the Spanish National Statistics Institute (INE), Alicante covers 5,820 km² with a population of 1,955,268; Castellón covers 6,637 km² with 604,086 inhabitants; and Valencia covers 10,810 km² with 2,656,841 inhabitants. Most of the population is concentrated along the relatively flat coastal strip, although certain mountainous coastal areas show significant development of tourist housing and second residences. In Fig. 3b, population densities for each municipality in 2021 are represented as the number of inhabitants per 100 m² of *GFA*. The distribution follows the expected pattern of higher values along the coast and lower values in the mountainous interior ranges.

The lithology of the mountainous zones in the study area is essentially carbonate (limestone and dolomites with substantial marl layers) from the Late Cretaceous period. The foothills consist of later Tertiary and Quaternary clay and silt deposits. The incidence of intense Alpine tectonics has resulted in a high degree of rock fracturing, which favours slope instability.

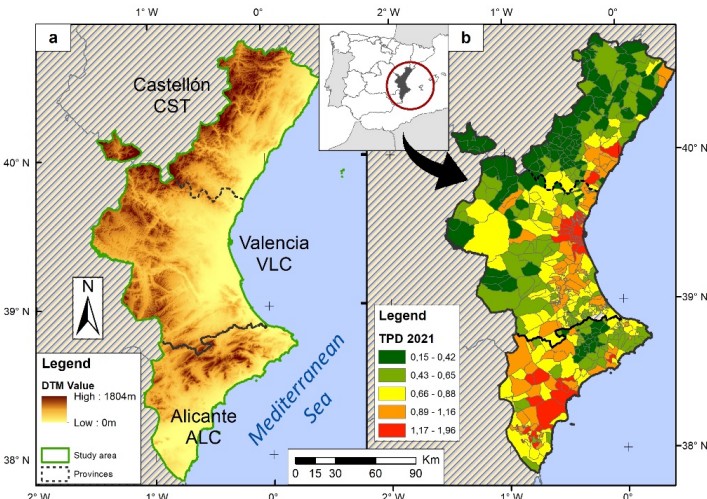

**Figure 3.** (a) physical map and (b) population density map (2021) expressed in inhabitants per 100 m² of GFA.

Urban development in the region is partially regulated by the *Land Use Planning and Landscape Protection Law* (LOTPP, 2021), whose article 15 defines the *Territorial Strategy of the Valencian Community (ETCV)* as the instrument that establishes objectives, criteria, and guidelines for territorial planning. This framework aims to limit the spread of mass tourism, which expanded throughout Spain after 1970 (Galiana Martín and Barrado Timón, 2006)). This trend has been particularly intense in the Valencian coastal zones, reaching its maximum expression along the northern coast of Alicante. As a result, these territories have experienced significant urban expansion (Gielen et al., 2018), often without adequately considering the impact of natural hazards.

### 3.2 Starting data

#### 3.2.1 Cadastral data

In Spain, local administration is organized into municipalities, and their residential surface data are recorded in the cadastral parcels that compose each municipality. These cadastral data were obtained from the *Cadastral Cartography Services* compliant with the INSPIRE Directive, provided by the Spanish General Directorate of Cadastre. The cadastral information, adapted to the European INSPIRE Directive, is available through interoperable services (WMS and WFS) and can be downloaded by municipality. Among the attributes provided by cadastral parcels (or *Dwelling Units, DUs*) are those required for this study: built surface (*GFA*), year of construction, and type of use. Accordingly, functional cadastral parcels with residential use were selected, while those with a construction date prior to 1900 were excluded. A first period between 1900 and 1950 was used to calculate an initial cumulative risk as a baseline. Subsequently, decadal series were defined beginning in 1950, which reduces random annual variability. The most recent decades coincide with the official census years in Spain (1981, 1991, etc.), produced by the Spanish National Statistics Institute (INE).



### 3.2.2 Valuation of residential buildings

The Spanish Cadastre provides an annual municipal property valuation, updated yearly in the *Official State Gazette*, based on a specific coefficient assigned to each municipality. However, this coefficient is not revised every year and varies widely, making it unsuitable for comparable values at a given moment. For this reason, cadastral valuations were not used. Instead, it was found more practical and realistic to use *Dwelling Value (DV)* expressed as the market appraisal price in €/m², available from real estate web portals. These statistics are based on second-hand housing sale listings published by private users and professional agents. Some portals consulted include RealAdvisor, Idealista, Fotocasa, and Hogaria.net. Publicly available municipal-level data were averaged to obtain *DV*. For each dwelling unit or cadastral parcel (*DU*), the value in euros was calculated by multiplying its surface area by the market appraisal (*DV, €/m²*) for each decade of the time series. Constant euros from the first quarter of 2022 were used, enabling comparisons of accumulated increases in housing stock without the influence of inflation.

### 3.2.3 Landslide inventory and databases

First, landslide mapping at 1:50,000 scale in vector format was used, produced by the *Regional Ministry of Public Works, Urbanism, and Transport (COPUT)* of the Generalitat Valenciana in the project *Lithology, industrial rock exploitation and landslide risk in the Valencian Community (Martinez and Balaguer 1998)*. This map was developed from geological and geotechnical data of the *Spanish Geological and Mining Institute (IGME)*, topographic maps at 1:50,000 scale, and aerial photographs available at that time. It was used to calculate spatial probability of landslides across susceptibility classes defined in LSMs. Additionally, the national-scale database of ground movements known as *BD-MOVES* (IGME) was used, which complies with the INSPIRE Directive (2007/2/EC). This database, created in 2014, constitutes Spain's national inventory of ground movements. It is structured into two georeferenced information blocks: (i) the description of intrinsic and relatively invariable characteristics of the movement, and (ii) the different activity events that generated such movements, including morphometry, triggering factors, and damages. This database was used to locate movements in the three provinces and, by considering their occurrence dates, to calculate their temporal probability.

### 3.2.4 Landslide susceptibility mapping

The susceptibility levels defined in the *Landslide Susceptibility Map (LSM)* developed in a previous study (Cantarino et al., 2019) were used. This mapping is characterized by a resolution of 25 × 25 m and the application of a *Spatial Multicriteria Evaluation (SCME)* method to weight the selected factors: slope, lithology, and land cover. Specifically, the susceptibility thresholds defined in that work were applied, particularly the medium, high, and very high classes (L3 to L5). These thresholds were derived through an objective and detailed classification based on ROC (*Receiver Operating Characterization*) analysis, which exploits the intrinsic variability of the data and represents one of the first applications of this type of maps. For this study, the spatial probability of each class was determined by comparing these susceptible areas with those recorded in the inventory. Combined with temporal probability, this enabled the calculation of hazard and, ultimately, risk assessment.

## 4 Method development

The methodology described in Section 2 was applied using the specific data of the study area presented in Section 3. Its calculation process was illustrated in the flowchart shown in Fig. 1.

### 4.1 Landslide risk calculation

As outlined in the methodology section, risk calculation requires evaluating the interaction between hazard (probability), exposure (value of the elements at risk), and vulnerability (severity of potential damage). In this case, since specific values are not available for each dwelling, risk was calculated for the entire local entity, in accordance with Eq. (2). The resulting risk values are expressed in euros (€) for the case study. The application of Eq. (2) to this particular case is detailed in the following subsections.

### 4.1.1 Hazard

To calculate the spatial and temporal probability of landslides, according to Eq. (3) in Section 2.1.1, the landslide databases described in Section 3.2.3 were used. The spatial probability ($P_s$) was calculated from the number of potential landslide areas, based on the COPUT landslide inventory for the study area, following Eq. (4). In this case, $P_s$ was determined for each susceptibility level considered (L3, L4, and L5), since L1 and L2 do not include representative landslides. *SRLi* was obtained from the COPUT inventory. An adjustment factor (*Faj*) of 20% was



applied, derived from the ratio between built surface within risk zones and total risk surface. The annual probability ($P_a$) was obtained using Eq. (5) with the BD-MOVES database of the IGME, which reports landslide events along with their location and date. Given the relatively low frequency of landslides in the study area, a decadal probability
($P_d$) was also computed. This value is derived from $P_a$ using the binomial (Bernoulli) model, whose probability mass function for $n$ trials and $k$ successes (each trial with probability $p$) is:

$$P(k,n,p) = \binom{n}{k} \times p^k (1-p)^{n-k} \tag{20}$$

Applying this to the probability of at least one event in a 10-year period yields:

$$P_d = 1 - P_a[k=0] = 1 - (1-P_a)^{10} \tag{21}$$

Using BD-MOVES, duplicate events and minor landslides ($\approx 300 \times 300$ m) were excluded, and landslides were classified by their location within each LSM level. In total, 73 landslides of sufficient size were identified across the three provinces, excluding variants of the same event. A 70-year interval was adopted for all levels, corresponding to the period with more systematic records, although earlier records exist. Since decadal probability is of interest, probabilities were recalculated using the binomial formulation in Eqs. (20) – (21). The two lowest
susceptibility levels (L1 and L2) were excluded (only included in the total SL column), as no landslides were recorded in these zones. In addition, the adjustment coefficient Gaj was not applied. The results are summarized in the following tables 6 and 7.

**Table 6.** Landslides, level, and risk surfaces.

| Level | Class | Nº landslides (CST/VLC/ALC) | SL Level Surface Area (km²) (CST/VLC/ALC) | SRL Risk Surface Area (km²) (CST/VLC/ALC) |
|---|---|---|---|---|
| L3 | Medium | 3 / 3 / 5 | 1986.8 / 3207.5 / 1717.8 | 9.0 / 19.0 / 7.2 |
| L4 | High | 8 / 6 / 10 | 1038.8 / 1628.0 / 639.8 | 17.2 / 83.3 / 28.3 |
| L5 | Very High | 13 / 12 / 13 | 394.9 / 681.4 / 439.7 | 18.2 / 36.9 / 23.9 |
| Total | | 24 / 21 / 28 | 6563.5 / 10690.7 / 5724.3 | 44.4 / 139.2 / 59.4 |

**Table 7.** Hazard calculation.

| Level | Class | Temp. Prob. (Pd (CST/VLC/ALC) | Spatial Prob. (Ps) (CST/VLC/ALC) | Hazard (H) (CST/VLC/ALC) |
|---|---|---|---|---|
| L3 | Medium | 0.075 / 0.075 / 0.125 | 0.005 / 0.006 / 0.004 | 0.0003 / 0.0004 / 0.0005 |
| L4 | High | 0.200 / 0.150 / 0.250 | 0.017 / 0.051 / 0.044 | 0.0033 / 0.0077 / 0.0111 |
| L5 | Very High | 0.325 / 0.300 / 0.325 | 0.046 / 0.054 / 0.054 | 0.0150 / 0.0163 / 0.0177 |

### 4.1.2 Exposure

In this study, only residential buildings are considered. These are characterized by relatively homogeneous typologies in the study area, particularly new vacation housing such as terraced or detached houses. Apartment blocks are generally not constructed in the areas under consideration, but rather in flat zones and/or near the coast.
Building height will be considered later in the calculation of vulnerability; however, for exposure a single appraisal value ($DV$) is applied to all constructions within the same municipality. The valuation of these elements was based on the habitable surface ($GFA$) provided by the Cadastre for each parcel, multiplied by the average municipal appraisal value obtained from real estate portals, as described in Section 3.2.2. Thus, exposure is expressed as the total value in euros per municipality for all dwellings exposed to landslide risk. The construction value used to
obtain exposure for each cadastral parcel or building exposed to potential landslides, according to Eq. (6), is given by:

$$E_{LE}(euro) = \sum GFA_{DU}(m^2) \times DV_{LE}(euro/m^2) \tag{22}$$



This metric makes it possible to estimate the reconstruction value of each parcel and municipality based on its built surface. All values are expressed in constant euros from the first quarter of 2022, as noted in Section 3.2.2.

### 4.1.3 Vulnerability

Vulnerability is a function of the magnitude or intensity of the landslide (*Landslide Magnitude, LM*) and the resistance capacity of the exposed element (see Section 2.1.3). These data are not necessarily included in the available landslide inventories (e.g., BD-MOVES and COPUT), although the depth of the planar slip surface typically ranges between 1 and 1.5 m (Pereira et al., 2012). The landslides occurring in the study area are shallow and of small magnitude, such that they do not completely destroy buildings. This type of building damage caused by landslides is classified by Léone (1996)as level III (on a scale from I to V), which corresponds to structural damage between 0.4 and 0.6 on a 0–1 scale. Considering this example and the fact that shallow slides in the study area show little variability in affected area, slip surface depth, velocity, volume, and typical damage, a single fixed value of LM was assumed [according to susceptibility level]. Therefore, LM was set at 0.6 on a heuristic scale ranging from 0 to 1 (Silva and Pereira, 2014).

The other factor in vulnerability assessment is the resistance of residential buildings (*Building Resistance, BR*), which depends on construction typology. In the study area, building techniques, materials (concrete), and structural types are relatively uniform and generally well preserved, so resistance is considered adequate. Construction age was not considered, assuming that most exposed buildings correspond to recent development, since safer areas had already been occupied. The main difference lies in the average number of floors per dwelling, although this is generally low and shows little variation. According to Section 2.1.3 and Eq. (7), higher BR values are assigned to taller buildings. Thus, the final vulnerability (*V*) depends only on the number of floors (*NF*), which was also obtained from the Spanish Cadastre. Table 8 presents the values used for this calculation, following a previous study of the area (Cantarino et al., 2021).

**Table 8.** Vulnerability as a function of building height.

| Number of floors (NF) | Landslide Magnitude (LM) | Building Resistance (BR) | Vulnerability (V) |
|---|---|---|---|
| > 8 | 0.6 | 30% | 0.42 |
| 8 – 4 | 0.6 | 20% | 0.48 |
| 4 – 2 | 0.6 | 10% | 0.54 |
| < 2 | 0.6 | 0% | 0.60 |

### 4.1.4 Final risk calculation

The final risk value for each local entity is obtained by applying Eq. (2) together with the formulations presented above. Thus, the total risk value for a given LE, according to its dwelling units (*DU*) located within a certain risk level (*i*), is expressed as:

$$RV_{LE}\,(\text{euro}) = \sum RV_{DU}\,(\text{euro}) = \sum [Ps_i \times Pd_i] \times [GFA_R \times DV_{LE}] \times [LM \times (1 - BR_{DU})] \tag{23}$$

It is important to stress that this study does not aim to produce an exhaustive or highly rigorous quantitative risk assessment. Rather, its main purpose is to provide a first approximation of the situation and evolution of landslide risk in each local entity, using comparative and relational procedures. Greater accuracy or complexity in risk calculation would not lead to significantly different outcomes, since the quantitative results are ultimately synthesized into only five management categories.

### 4.2 Risk assessment

To obtain an objective evaluation of the state and evolution of landslide risk, the indicators defined in the methodology were calculated. Specifically, values of *RI*, *RSI*, and *RQI* were obtained for all decades in the available series, beginning in 1950/60, which coincides with a notable increase in construction activity resulting from the country's economic and tourism boom. The trend in risk growth (*mRQI*) was calculated as the slope of the regression line (expressed in sexagesimal degrees) for the four most recent decades, i.e., between 1981 and 2021. The 1950/60 and 1960/70 decades were excluded due to their large variations—linked to the aforementioned surge in construction activity—and to avoid working with an excessively long series. As justified in the methodology, the trend in risk surface growth (*mRSI*) was not included. Finally, risk classification was carried out for the analyzed municipalities. The quantiles defined in Section 2.2.2 were applied to establish five classes for



each province separately, thus individualizing results. The threshold distinguishing the two RSI levels was set at
the P60 percentile. The 2011/21 decade was used for state variables, while the 1981–2021 interval was used for
growth variables. For percentile calculations, the 260 municipalities with relevant risk (*RI > 0.001*) were selected.
Additionally, provincial differences were identified, which are discussed in the Results section.

Table 9 presents the percentiles used to define the lower limits of the index levels. The province of
Castellón shows higher quantile thresholds for the state variables, which appears to indicate a greater risk situation
due to its mountainous orography and the predominance of small settlements. This confirms the interpretation
noted in Table 1. However, the trend values are similar to those of the province of Alicante, where larger
populations and more intense construction activity have driven comparable growth rates.

**Table 9.** Quantile values for index thresholds.

| Percentile | Castellón (RSI / RQI / mRQI) | Valencia (RSI / RQI / mRQI) | Alicante (RSI / RQI / mRQI) |
|---|---|---|---|
| P90 | 0.91 / 1.10 / 12.78 | 0.65 / 0.58 / 8.73 | 0.68 / 0.46 / 12.63 |
| P60 | 0.35 / 0.36 / 0.05 | 0.19 / 0.29 / 0.02 | 0.31 / 0.24 / 1.03 |
| P40 | 0.15 / 0.21 / –0.54 | 0.10 / 0.19 / –1.32 | 0.15 / 0.16 / –1.11 |
| P20 | 0.06 / 0.11 / –8.24 | 0.04 / 0.09 / –4.95 | 0.08 / 0.08 / –4.36 |

## 5 Results

As a result of the first part of the methodology, all indices were calculated for the 542 municipalities of the
490 Valencian Community across the decadal series beginning in 1950. The average values of the state indices *RSI*
and *RQI* for municipalities with significant risk are shown in Fig. 4. This graph shows a fairly stable evolution
with a slight downward trend, with some nuances. The province of Castellón stands out due to its higher values,
explained by its mountainous terrain and the prevalence of smaller municipalities where higher index values are
reached. A slight increase in *RSI* is also observed in Alicante, driven by its role as a vacation destination in coastal
municipalities. However, this rise is not accompanied by an increase in *RQI*, indicating lower occupation of high-
susceptibility zones. Subsequently, *mRQI* values were calculated for the 1981–2021 series. The averages (Alicante:
–0.02°; Castellón: 1.03°; Valencia: –0.40°) again highlight the higher values of Castellón, consistent with the
reasons described above. Using all these data, the 542 municipalities were classified according to the management
types defined in the methodology. This provincial breakdown is presented in Table 10.

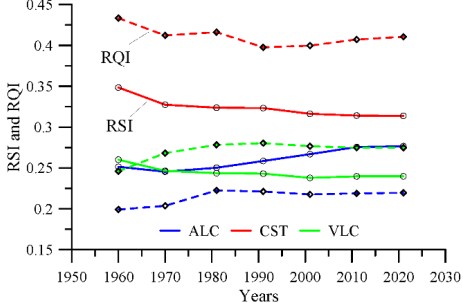

**Figure 4**. Decadal evolution of state indices.

Table 10 shows that only 22 municipalities exhibit a clear need to improve their risk management.
Castellón stands out with 9 cases, despite not being the largest province, as many small municipalities are located
in mountainous areas where high index values are more easily reached (see Section 4.2). At the other extreme, 282
municipalities show no significant exposure to landslide risk. Castellón and Alicante show similar results but under
505 very different scenarios. When population density (see Fig. 3b) is considered, the nine Castellón municipalities
present an average of 0.41 inhabitants/100 m² GFA, whereas the six municipalities in Alicante show an average
of 0.80. With nearly double the density, the higher index values in Alicante result from stronger construction
activity driven by housing demand pressures. These municipalities are larger, more populated, and benefit from
greater economic development, with tourism playing a decisive role.





**Table 10.** Management types by municipality.

| Province | Very Improvable | Improvable | Reviewable | Suitable | Unaffected | TOTAL |
|---|---|---|---|---|---|---|
| Castellón | 9 | 13 | 14 | 31 | 68 | 135 |
| Valencia | 7 | 30 | 29 | 46 | 154 | 266 |
| Alicante | 6 | 24 | 23 | 26 | 62 | 141 |
| TOTAL | 22 | 68 | 65 | 105 | 282 | 542 |

Figure 5 presents four maps depicting, for all municipalities, the variables most influential in defining
risk management, according to the quantiles specified in Table 9. Panel 5d summarizes the overall management
classification. The figure illustrates the spatial distribution of the different indices. High *RSI* values are
concentrated mainly in inland areas, whereas high *RQI* values also extend to coastal zones, a pattern further
accentuated in *mRQI*. The classification map (5d) is particularly informative, showing municipalities with higher
management levels in inland Castellón, while in Alicante they are predominantly located along the coastal strip.

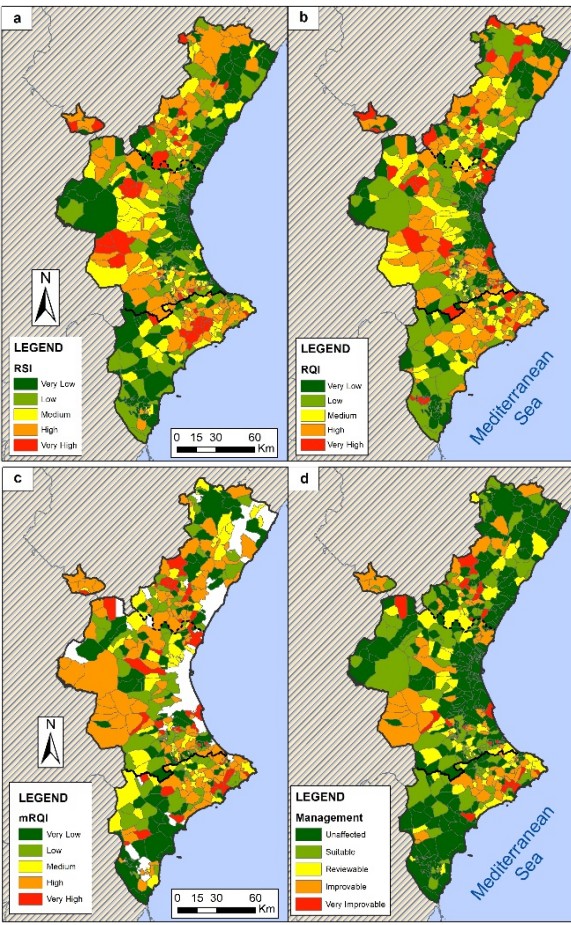

**Figure 5.** Municipality classification: (a) RSI; (b) RQI; (c) mRQI and (d) management type.

As noted, some municipalities historically located in risk areas show an *RSI* value of 1 throughout the
series (GFA = GFA$_R$). Based on the processed data, these maximum values occur in seven municipalities in
Alicante and four in Castellón, all of them small, mountainous settlements where 100% of the built surface lies in
medium-to-high susceptibility zones. This problem is linked to the siting of historic town centers, although no
landslides have occurred within urban areas over the last century. This confirms that the indicator reflects exposure
quantity rather than risk quality.



With respect to *RQI*, 17 municipalities in Castellón and 3 in Valencia exceed the value of 1. This indicates that most of their risk surface lies above level L4 (see Table 1), a situation only possible in small municipalities. In all these cases, this is explained by the partial location of historic town centers in high-susceptibility zones. Only six municipalities show a significant increase in the temporal series, all in Castellón, likely reflecting demand for second homes, since these are not major tourist destinations. High *RQI* values (>0.7) may coincide with low *RSI* values, a relatively common situation in small settlements. This indicates that while the exposed surface is small, it is largely situated in high or very high susceptibility zones. Since the affected surface is limited (RSI "b") and *RQI* is not high, their management types are classified as Reviewable or Suitable.

Noteworthy is the concentration of "Very Improvable" municipalities along the coastal strip of Alicante, encompassing well-known vacation destinations with high population density. This includes the La Marina region, which was previously studied by the same authors (Cantarino et al., 2021), with results consistent with those presented here. Specifically, the municipalities of Altea and Benitatxell, already identified in the earlier study, again emerge in this work, although the methodological approach differs.

According to Fig. 5d, it is also significant that municipalities in the coastal mountain ranges of Castellón, many of them subject to tourism pressures similar to those in Valencia (Cullera) or Alicante (La Marina), are not classified as highly affected. This is due to the fact that these Castellón ranges are protected by the Valencian regional government as natural parks, thus avoiding residential urban expansion into unsuitable areas.

## 6 Discussion

This section provides a global perspective of the results for the 260 municipalities with significant landslide risk (RI > 0.001). First, it analyzes the relationships among the main municipal variables obtained during the course of the study. Second, it examines the evolution of the key variables used in this research through the definition of a growth rate that enables straightforward graphical comparison. The supplementary municipal variables, not explicitly described earlier, are presented in Table 11.

**Table 11.** Supplementary LE variables.

| Variable | Units | Description |
|---|---|---|
| SpGFA | ha | Specific rate of construction activity in the municipality, calculated as GFA (m²) relative to the total municipal surface area for 2021 |
| mSpGFA | degrees | Trend in total construction activity between 1981–2021 |
| PopT | inhabitants | Total population per municipality for 2021 |

To examine the relationship of these supplementary variables together with DV, RI, RQI, and RSI (for the year 2021, across 260 municipalities), a correlation analysis was conducted, highlighting the main results. The correlation matrix obtained reveals a statistically significant and positive correlation between the appraisal value (DV), population (PopT), and construction activity trend (mSpGFA). This is a logical result reflecting normal growth. However, although there is also a significant correlation between RI and RSI, as well as between population and construction activity, the sign is negative, with a stronger effect for RSI. In other words, in larger municipalities with intense construction activity, RI values are lower and, in particular, the share of built surface located in risk zones decreases. These indices, including RQI, show no significant correlation with other variables such as property appraisal value, which can be explained by the fact that the highest risk values do not occur in the most touristic settlements.

Figure 6 provides a summary, by province and decade, of risk value RV, GFA and population for municipalities with significant risk (*RI > 0.001*). In addition, a set of risk indicators based on the concept of growth rate (*GwR*) or variation is proposed:

$$GwR = \frac{(Final\ value - Initial\ value)}{Initial\ value} \times 100 \tag{24}$$

In all cases, a significant increase in construction activity (80%) is observed between the period of economic and tourism takeoff (1960s/70s) and 1981. Thereafter, the trend gradually moderates. The low construction rate of 2011–21 clearly reflects the impact of the economic crisis. Risk values stabilized from the 1970s, when the first phase of construction expansion had ended. Importantly, this process does not parallel population growth, suggesting that construction activity was driven less by housing demand and more by tourism or second homes.

The province of Alicante stands out in all growth indicators, particularly those linked to population dynamics, despite its rugged orography. Its construction activity is much greater, especially in terms of risk values, but not proportional to its population dynamics, indicating that a significant share of development serves as



secondary residential destinations. At the opposite end, Castellón shows a more rural profile, with lower population levels and some construction in risk zones, but with lower absolute values. This suggests a less touristic profile, with development more related to primary housing and second homes. Therefore, it is essential to combine population data with risk metrics to obtain a complete picture of each area under study. Population data alone cannot resolve the issue of second homes and their occupancy rate, but knowing the magnitude of the affected population is a dimension of risk that must be studied with the same intensity as housing exposure itself.

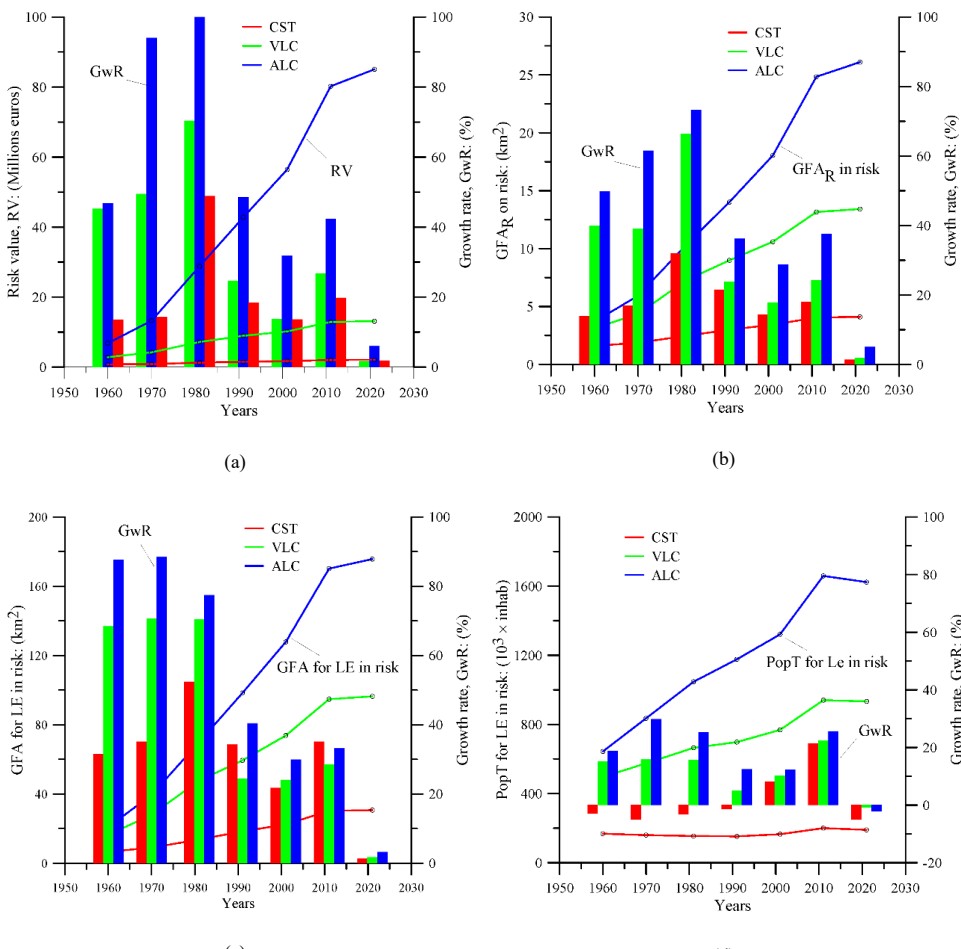

**Figure 6.** Historical evolution of some variables (lines) and their growth rate (GwR, columns): (a) Risk value (RV, million €, 2021); (b) GFA in risk (GFA$_R$, km²); (c) Total GFA for LE in risk (km²); (d) Population for LE in risk (PopT, $10^3$ x inhabitants).

## 7 Conclusions

A key outcome of this study is the objective assessment of landslide risk across a broad set of local entities or municipalities through the definition of a series of indices that are straightforward to compute once risk valuation is performed. The indices that most clearly explain relative risk conditions are those related to *risk quality*, i.e., those based on the proportion of L4 and L5 zones within the risk surface. Logically, these are the areas with the highest probability of landslide occurrence. Particularly noteworthy is the trend value of the quality index (*mRQI*), which defines the evolution of high risk in a given local entity and, when increasing, calls for reflection on territorial management practices. In other words, mRQI indicates the adequacy of municipal construction dynamics, highlighting potential progressive occupation of unsuitable land for residential housing in high-risk zones.



Secondly, in the case of the Valencian Community, it is noteworthy that, despite an increase in the total risk-prone surface, no clear rise in risk indices is observed across the analyzed period. This stability indicates that risk levels have not grown in proportion to overall construction, suggesting a slowdown—though not a complete halt—in the occupation of unsuitable areas for residential purposes.

At the provincial level, Castellón stands out with nine municipalities that require urban planning review. As previously noted, these are all small, sparsely populated localities in mountainous inland areas that have faced high risk since the beginning of the study period (see Fig. 5d). By contrast, the seven municipalities in Alicante requiring revision are located along the densely populated coastal strip and are characterized by significant concentrations of vacation housing (Fig. 5d).

However, no direct link has been identified between tourism-oriented urbanization and the occupation of unstable zones. Tourism does not appear to be the primary driver, although it plays a notable role in certain coastal municipalities. In general, the affected areas are mountain localities somewhat removed from the coast that have experienced progressive urbanization, driven less by tourism than by residential demand.

Indeed, population dynamics do not appear to explain the urban expansion observed in these rural
municipalities, as increases in inhabitants are not synchronous with housing development. Instead, the expansion seems largely driven by second homes used as vacation residences by families with ties to these areas. In this context, incorporating the geolocation of populations exposed to landslide risk would be appropriate, as it would complement the current analysis, which considers only the economic value of affected dwellings.

Ultimately, this study presents a detailed comparative analysis of an extensive territory using a consistent
calculation procedure, thereby providing an objective tool for land-use managers. The results are robust, as they rely on a sizeable sample of 275 municipalities distributed across three provinces with distinct dynamics. These findings demonstrate the replicability of the methodology in other landslide-prone areas where comparable baseline data are available.

Forecasting studies of natural hazards are essential for mitigating their impacts. Although government
administrations generally consider them, this is often done only partially. These studies must be progressively refined to provide a robust and comprehensive tool for territorial management. Only in this way will society be able to confront the growing significance of such hazards and, if not prevent them, at least mitigate their severe consequences—such as the extraordinary floods that struck our region at the end of 2024, causing more than two hundred fatalities and extensive material losses.


### Data availability

Data supporting the findings of this study are available from the authors upon request.

### Authors contribution

I.C.: Conceptualization, Methodology, Software, Validation, Formal Analysis, Investigation, Data Curation,
Visualization, Writing (original Draft); M.A.C.C.: Methodology, Validation, Writing (review and editing) Visualization; E.G.: Methodology, Validation, Writing (review), Visualization; J.S.P.J.: Methodology, Validation, Writing (review), Visualization.

### Competing interest

The authors declare that they have no competing interests.

### 635 Acknowledgement

This research received no specific grant from any funding agency in the public, commercial, or not-for-profit sectors. The authors gratefully acknowledge the Universitat Politècnica de València (UPV) for providing access to the software used in this study.

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
