# Peer review of "Spatiotemporal assessment of landslide risk over large areas: A case study of the Valencian Community (1950–2021)"

_EGUsphere, 2025_

## Author Comment (AC1)

**RESPONSE TO REVIEWER 1**

We thank the reviewer for the constructive comments and suggestions, which have helped to improve the quality and clarity of the manuscript. Below, we provide a point-by-point response to each comment. In the following pages, text in italics indicates our responses to the reviewer, while changes and modifications made to the manuscript are indicated in quotation marks and in red.

**1. Major comments**

The manuscript frequently claims that the proposed framework is particularly suitable for large, spatially heterogeneous regions such as the Valencian Community, yet it does not clearly explain why such regions are challenging for landslide risk assessment or how the proposed methodology addresses those challenges. Large and heterogeneous areas are typically characterized by strong geological and geomorphological variability, uneven spatial distribution of landslide inventories, inconsistent resolution of socioeconomic data, and mismatch between geological and administrative boundaries - all of which can undermine comparability and accuracy of regional risk models. The authors should explicitly discuss these challenges and clarify how the RQI, RSI, and mRQI frameworks overcome them, for instance by integrating multi-source datasets, applying normalization to reduce bias among different scales, coupling physical and socioeconomic factors across scales, and using dynamic indicators to capture temporal evolution of risk. Expanding this discussion would convincingly demonstrate the framework's innovation and justify its claimed applicability to spatially dispersed regions.

*We thank the reviewer for this insightful comment. In response, we have revised the manuscript to explicitly explain why large, spatially heterogeneous regions pose specific challenges for landslide risk assessment and to clarify how the proposed framework addresses these challenges.*

*First, we have expanded the methodological section by adding a new subsection (Section 2.3) that explicitly discusses the main sources of heterogeneity affecting regional-scale landslide risk assessment, including geological and geomorphological variability, uneven spatial coverage of landslide inventories, differences in spatial resolution among datasets, and mismatches between physical processes and administrative boundaries. This section clarifies how the proposed framework integrates multi-source datasets within a coherent spatial reference system and applies normalization procedures to ensure comparability across scales.*

*Second, the description of the study area (Section 3.1) has been revised to better emphasize the spatial, geomorphological, and socio-economic diversity of the Valencian Community, as well as the historical patterns of dispersed and fragmented urban development that complicate exposure and risk assessment at the regional scale.*

*Finally, the Conclusions (Section 7) have been reinforced to explicitly highlight the suitability of the proposed framework for large and heterogeneous regions, emphasizing the role of dimensionless risk indices and dynamic trend indicators (mRQI, mRSI) in capturing both spatial contrasts and temporal evolution of landslide risk.*

*These revisions collectively strengthen the methodological justification of the framework and more clearly demonstrate its applicability to large, spatially dispersed regions, as suggested by the reviewer.*

*To begin with, we have added a new Section 2.3 explicitly addressing these challenges*

*"**2.3. Considerations on data sources and regional heterogeneity***

*Large and spatially heterogeneous territories, such as the Valencian Community, pose significant challenges for landslide risk assessment due to pronounced geological and land-use variability, uneven and incomplete spatial coverage of landslide inventories, and differences in the spatial resolution of available datasets. In addition, the lack of correspondence between physical variables and the administrative units used in socioeconomic datasets can hinder data aggregation and limit the comparability of risk estimates across scales.*

*To address these limitations, the proposed methodological framework integrates multi-source datasets within a consistent spatial reference system and applies normalization procedures to ensure that indicators derived from different units and spatial resolutions are directly comparable. Landslide susceptibility maps incorporate geological variables, digital elevation models (DEMs), and land-use information, which are combined through weighted coefficients derived from multicriteria evaluation methods. Risk calculation is further supported by the use of residential building economic value as a proxy for exposed wealth.*

*The methodological design also explicitly addresses regional heterogeneity by linking the risk surface index (RSI, representing the extent of built-up areas under risk) with the risk quality index (RQI, representing risk intensity) within the global risk index (RI). This coupling allows the framework to capture spatial contrasts between mountainous areas prone to landslides and densely built residential zones, ensuring a coherent representation of risk across the entire territory. In addition, the temporal trend indicator of risk quality (mRQI) provides a dynamic measure of changes in exposure and vulnerability, preserving both the magnitude and direction of change through normalization to a –1 to +1 scale.*

*Although the present implementation focuses on the Valencian Community as a case study, the proposed framework is designed to address challenges inherent to large and heterogeneous regions. Future applications may further refine this approach through the systematic incorporation of additional normalized variables and extended temporal series, enhancing its transferability and robustness in diverse territorial contexts."*

*In the Section 3.1 (Study Area), we have modified the last paragraph (lines 330-337) by:*

*"The Valencian Community is not only extensive but also highly heterogeneous, comprising coastal metropolitan areas, intermediate zones, and mountainous inland regions with distinct geomorphology, land-use patterns, and socio-economic dynamics (Gielen et al., 2018). This diversity poses a major challenge for landslide risk assessment because susceptibility factors, exposure, and vulnerability vary significantly across municipalities. Furthermore, urban development has historically followed a dispersed and discontinuous pattern (Gielen et al., 2018), particularly in coastal and peri-urban areas, which increases the complexity of mapping exposure and evaluating risk. These characteristics require a methodology capable of integrating detailed cadastral data and susceptibility mapping at a fine spatial scale, ensuring comparability across such a fragmented territory.*

*Although urban development in the region is partially regulated by the Land Use Planning and Landscape Protection Law (LOTPP, 2021), whose article 15 defines the Territorial Strategy of the Valencian Community (ETCV) as the instrument for territorial planning, this framework was introduced relatively late. Its aim is to limit the spread of mass tourism, which expanded throughout Spain after 1970 (Galiana Martín and Barrado Timón, 2006). This trend has been particularly intense in the Valencian coastal zones, reaching its maximum expression along the northern coast of Alicante. As a result, these territories have experienced significant urban expansion (Gielen et al., 2018), often without*

*adequately considering the impact of natural hazards—further reinforcing the need for a risk assessment approach adapted to large, spatially heterogeneous regions."*

*In section 7 (Conclusions), the first paragraph (587-595 lines) has been replaced by:*

*"The proposed framework demonstrates its suitability for large, spatially heterogeneous regions by addressing two critical issues: territorial diversity and fragmented urbanization. As highlighted by Gielen et al. (2018), the Valencian Community has experienced significant urban sprawl, resulting in scattered residential areas often located in zones with varying hazard levels. By incorporating high-resolution cadastral data and defining dimensionless indices (RI, RSI, RQI) together with trend indicators (mRQI, mRSI), our approach enables consistent evaluation of both the magnitude and quality of risk across municipalities, in contrast to traditional approaches that rely on regional averages and assume territorial homogeneity. Particularly noteworthy is the trend value of the quality index (mRQI), which defines the evolution of high risk in a given local entity and, when increasing, calls for reflection on territorial management practices. In other words, mRQI indicates the adequacy of municipal construction dynamics, highlighting potential progressive occupation of unsuitable land for residential housing in high-risk zones."*

*As a final point, regarding the normalization of the risk indices, we would like to clarify that RSI and RQI are already defined as dimensionless indicators within the 0–1 range. Following the reviewer's suggestion, we have therefore revised the only indicator that originally lay outside this range, namely the mRQI. This index is now expressed using a consistent, dimensionless normalization.*

*Accordingly, the mRQI values reported in Table 9 have been normalized to the 0–1 range, improving their interpretability and facilitating direct comparison across municipalities and time periods, without altering the relative trends captured by the index.*

**Table 9.** Quantile values for index thresholds, with mRQI expressed in normalized values.

| Quantil | Castellón | | | Valencia | | | Alicante | | |
|---|---|---|---|---|---|---|---|---|---|
| | RSI | RQI | mRQI | RSI | RQI | mRQI | RSI | RQI | mRQI |
| Q90 | 0,91 | 1,10 | 0,23 | 0,65 | 0,58 | 0,16 | 0,68 | 0,46 | 0,23 |
| Q60 | 0,35 | 0,36 | 0,00 | 0,19 | 0,29 | 0,00 | 0,31 | 0,24 | 0,02 |
| Q40 | 0,15 | 0,21 | -0,01 | 0,10 | 0,19 | -0,02 | 0,15 | 0,16 | -0,02 |
| Q20 | 0,06 | 0,11 | -0,15 | 0,04 | 0,09 | -0,09 | 0,08 | 0,08 | -0,08 |

**2. Minor comments**

**1.** The methodological innovation of the RQI, RSI, and mRQI indices relative to existing risk assessment frameworks should be stated more clearly. A concise comparison with earlier studies (e.g., Guzzetti et al., 2005; Pereira et al., 2020; Segoni & Caleca, 2021) would help readers understand the conceptual advancement of this work.

[New reference: Guzzetti, F.; Reichenbach, P.; Cardinali, M.; Galli, M.; Ardizzone, F. (2005). Probabilistic landslide hazard assessment at the basin scale. Geomorphology, 72(1-4), 272-299. DOI:10.1016/j.geomorph.2005.06.002.]

*We thank the reviewer for this helpful suggestion. Following this comment, we have incorporated the work of Guzzetti et al. (2005) and strengthened the description of the methodological innovation of the proposed indices (RQI, RSI, mRQI, and mRSI) relative to previous landslide risk assessment frameworks.*

*In the revised manuscript, the introductory paragraph of Section 2.2 ("Risk indices and qualifiers") has been modified to explicitly compare the proposed approach with earlier studies, including Guzzetti et al. (2005), Pereira et al. (2020), and Segoni and Caleca (2021). This comparison highlights the main conceptual advances of our framework, particularly the explicit decomposition of risk components, the use of dimensionless and comparable indices, and the incorporation of a temporal dimension through trend indicators.*

*These revisions clarify how the proposed indices build upon and extend existing methodologies, and more clearly position the contribution of this work within the current landslide risk assessment literature.*

*In this way, the methodological innovation of the proposed indices is now presented in a clearer and more comparative manner. The text replaced the existing paragraph (lines 180 to 188) at the beginning of Section 2.2. is as follows:*

*"Compared with previous methodologies, the risk indices proposed in this study incorporate several conceptual and technical innovations that enable a more flexible, reproducible, and comparative assessment of landslide risk at regional scale. In this framework, all residential buildings are considered at the Dwelling Unit (DU) level, together with their reconstruction value (DV) and susceptibility level ($L_n$), allowing for a more detailed quantification of risk and the analysis of its temporal evolution.*

*Unlike the static probabilistic hazard-focused approach proposed by Guzzetti et al. (2005), which was applied at the basin scale, the indices introduced here explicitly integrate a spatiotemporal perspective, enabling the assessment of risk evolution over an extended historical period (1950–2021).*

*Pereira et al. (2020) developed a Landslide Risk Index (LRI) at the municipal scale for the entire Portuguese territory by combining susceptibility and exposure factors through fixed empirical weights and applying a normalization procedure to facilitate inter-municipal comparison. While this approach allows spatial comparison, it does not explicitly address the temporal dynamics of risk.*

*Similarly, Segoni and Caleca (2021) proposed aggregated indicators of landslide risk (ALR and TLR) at the national scale for Italy by combining susceptibility and land consumption data. Their approach focuses on spatial aggregation for a single reference year and does not incorporate temporal trends or building-level economic exposure.*

*In contrast, the present framework decomposes landslide risk into complementary quantity (RSI) and quality (RQI) components and introduces trend indicators (mRQI, mRSI) that capture the direction and magnitude of temporal change. This design enables consistent spatial and temporal comparison across heterogeneous administrative units and represents a key methodological advancement over existing large-scale landslide risk assessment approaches."*

**2.** The relationships and calculation logic of the indices-particularly the variables Gaj, Faj, LM, and DV-need clearer description. Including a schematic diagram summarizing index derivation, weighting, and normalization would enhance transparency and reproducibility.

*We thank the reviewer for pointing out the need to clarify the relationships and calculation logic of the indices. In response, we have revised the manuscript to provide clearer and more explicit definitions of the variables Faj, Gaj, LM, and DV at their first occurrence in the text.*

*Specifically, the following clarifications have been incorporated:*

• **Faj (adjustment factor)** *accounts for the proportion of susceptible area that effectively affects inhabited zones. It is defined as the ratio between the area susceptible to landslides impacting populated areas and the total susceptible area within a given territory. Its value ranges between 0 and 1, and a representative average value of 0.2 was adopted based on the observed relationship between these surfaces.*

• **Gaj (adjustment factor)** *reflects the size or volume of the displaced material associated with the inventoried landslides and also ranges between 0 and 1. In the present case study, this factor was ultimately not applied, as the landslides considered exhibit similar magnitudes and volumes, making additional weighting unnecessary.*

• **DV (Dwelling Unit Value)** *represents the reconstruction cost of residential buildings, expressed in economic units per unit of built surface area (€/m²). The values used in this study were derived from recognized real estate agencies and are directly interpretable, and therefore were not subject to normalization.*

• **LM (Landslide Magnitude)** *expresses the intensity of the landslide in terms of size and impact energy and is commonly used in vulnerability assessments. Given the relatively homogeneous characteristics of shallow landslides in the study area, a single fixed value of LM = 0.6 was assumed on a heuristic scale ranging from 0 to 1, following Silva and Pereira (2014) and Cantarino (2021).*

*These revised definitions have been explicitly included in the manuscript at the lines where each variable is first introduced (Lines 147, 152, 163, and 169), improving the transparency and reproducibility of the index calculations.*

*Line 147 – "The adjustment factor (Faj) accounts for whether the event occurs in an inhabited area, derived from the ratio between the area susceptible to landslides affecting populated areas and the total susceptible area."*

*Line 152 – "This annual probability (Pa) must be adjusted using the factor Gaj, which reflects the volume of the displaced material in the inventoried landslides."*

*Line 163 – "In addition, a property valuation can be used to calculate the reconstruction cost of the dwelling, or Dwelling Unit Value (DV), expressed in economic units per unit of surface area, must also be considered."*

*Line 169 – "Vulnerability is a function of the magnitude or intensity of the landslide (Landslide Magnitude, LM) in energy and size terms. Depends on the resistance capacity of the affected element, which is closely related to building height."*

*The values of these indices did not need to be weighted as they are directly applicable. Furthermore, their very definition includes a normalised variation, excluding DV. However, DV is a recognised value that is easy to interpret and, in our opinion, does not require normalisation.*

*Regarding the suggestion to include a schematic diagram, we agree that such visual tools can be useful. However, given the relatively straightforward formulation of the indices and the number of variables involved, we consider that adding a diagram would substantially increase the length and complexity of the methodological section without providing proportional additional clarity. We therefore have opted to improve the textual explanations while keeping the methodology concise and readable.*

**3.** The study would benefit from a brief uncertainty or sensitivity analysis to evaluate how variations in data inputs (e.g., susceptibility classification or economic valuation) affect risk index results. Even a qualitative discussion would strengthen confidence in the robustness of the findings.

*We appreciate the reviewer's suggestion. The susceptibility maps used in this study were produced in previous official assessments, and the economic valuation of dwellings was obtained from external specialised organisations. The extreme input values and their implications are described in the manuscript.*

*With regard to sensitivity, the risk indices show consistent behaviour when applied to nearly 300 municipalities with appreciable risk levels. The use of percentile-based classification also contributes to standardising the results and reducing the influence of outliers. Moreover, the provincial analyses illustrate how different initial conditions and data ranges affect the final risk values, without revealing disproportionate variations or unstable intervals.*

**4.** Although the dataset spans more than seven decades, temporal changes in landslide risk are not well illustrated. Incorporating a time-series trend, decade-based comparison, or discussion of major shifts in risk drivers would make the "spatiotemporal" aspect of the study more convincing.

*We appreciate the reviewer's comment. The temporal variation is captured through the mRQI and mRSI indices, which quantify changes in landslide risk during the 1981–2021 period. This interval corresponds to the decades in which construction activity intensified in the region. As shown in Table 9, at least 10% of the municipalities (P90) exhibit a marked upward trend in risk levels.*

*We have incorporated a brief paragraph in the manuscript to clarify these temporal patterns and reinforce the spatiotemporal interpretation of the results, replacing the paragraph between lines 483 and 485:*

*"Table 9 presents the percentiles used to define the lower limits of the index levels. The province of Castellón shows higher quantile thresholds for the state variables, which appears to indicate a greater risk situation due to its mountainous orography and the predominance of small settlements. This confirms the interpretation noted in Table 1.*

*Moreover, the temporal indices reveal that at least 10% of the municipalities (P90) show a notable upward trend in risk during 1981–2021. Castellón and Alicante both display similarly elevated values, although driven by different factors: Castellón's increase is mainly attributable to its inherently more susceptible geomorphological conditions, while in Alicante the rise relates to tourism-driven development and growing construction pressure in areas exposed to slope instability.*"

**5.** The discussion of socioeconomic influences such as tourism and urban expansion remains qualitative. Integrating basic quantitative indicators-such as land - use change, population growth, or infrastructure density - would provide stronger empirical support for the interpretation.

*We appreciate the reviewer's insightful observation. Socioeconomic drivers such as tourism pressure and urban expansion indeed contribute to increased landslide risk. These influences are incorporated indirectly in our analysis through the economic valuation of dwellings (DV), which reflects local demand dynamics and the upward pressure exerted by tourism-intensive or rapidly developing areas. As a result, variations in DV partly account for underlying socioeconomic dynamics without introducing additional independent indicators.*

*Explicitly quantifying these drivers using variables such as land-use change, population growth or infrastructure density would certainly strengthen the socioeconomic interpretation; however, integrating such datasets would require a broader methodological framework and constitutes a substantial extension of the current study. As indicated in the 'Major comments' section, this represents a natural direction for future work, where socioeconomic indicators could be incorporated systematically into the risk model.*

**6.** The manuscript would benefit from editorial refinement. Ensure consistent terminology throughout (e.g., unify "risk zone," "susceptibility zone," and "management class"), verify incomplete references, and standardize equation formatting and figure captions for clarity and professionalism.

*We thank the reviewer for this observation. A thorough editorial revision of the manuscript has been performed. Terminology has been standardised throughout the text; specifically, the expressions 'susceptibility zones' have been replaced by 'risk zones' in lines 496, 525, 530 and 534, and the term 'risk zone' is now used consistently across the manuscript.*

*The expression 'management class' was not employed in this study; however, the term 'management classification', which is used in the manuscript, accurately reflects the categories defined in our methodology.*

*All equations have been reformatted using MathType to ensure consistency, the reference list has been checked and cross-verified through Mendeley, and all figures have been reviewed and standardised using Grapher to improve clarity and presentation quality.*

---

## Author Comment (AC2)

**RESPONSE TO REVIEWER 2**

We thank the reviewer for the constructive comments and suggestions, which have helped to improve the quality and clarity of the manuscript. Below, we provide a point-by-point response to each comment. In the following pages, text in italics indicates our responses to the reviewer, while changes and modifications made to the manuscript are indicated in quotation marks and in red.

**1. Major comments**

While this study does provide an interesting framework for calculating a (decomposed) risk index that enables comparisons across several administrative units and for the same unit over time, the paper, as it stands right now, seems more like an application of the methodology, without a proper evaluation of the strengths and limitations of the method, its implications for the research domain, or a clear indication of how this paper builds on existing research in the field. This is especially the case for the discussion and conclusion sections, which are too tailored toward the interpretation of the case study, without a proper evaluation of what this new index actually means for research and how it could be adopted beyond the case study. Some interesting questions to address would be: how would this framework perform in regions with limited compliance with building standards? Can we consider this approach applicable in such contexts? In which cases would it be applicable, and in which would it not? For instance, the paper mentions in the introduction that proper planning (and thus understanding risk) is especially relevant in developing countries, but from my understanding, the framework developed in the paper would be much more adequate in more "formal" urban contexts (as in the case study in Spain) than in complex, spontaneous settlements in the "developing world." It may be interesting to address this from a research perspective, as a separate subsection or part of the discussion, to clarify what this research adds and how it generalizes.

*We agree that the original version of the manuscript placed excessive emphasis on the case study application, while the broader methodological implications of the proposed framework were not sufficiently discussed. To address this concern, we have substantially expanded the Discussion section by adding a new subsection entitled* "6.2 Strengths, limitations and transferability of the proposed framework." *In this subsection, we explicitly evaluate the methodological contributions of the framework, discuss its main strengths and limitations, and clarify the contexts in which it is most applicable.*

*In particular, we now discuss the framework's suitability for regions characterized by formal urban development and reliable socioeconomic datasets, as well as the challenges associated with its application in contexts dominated by informal settlements or limited data availability. We emphasize that the framework should be understood as a flexible methodological structure that can be adapted to different settings, rather than as a universally applicable model.*

*Now, we have reorganized the Discussion into two subsections to clearly separate the interpretation of the case study results from the methodological implications of the proposed framework. We have also included some modifications taking into account the 'Minor comments'.*

**"6. Discussion**
**6.1. Interpretation of landslide risk in the Valencian Community**
 *- - - The original text remains the same - - -*

**6.2. Strengths, limitations and transferability of the proposed framework**

*The proposed framework represents a structured and transparent approach for assessing landslide risk over large and heterogeneous regions by decomposing risk into physically based susceptibility (RSI) and socioeconomically driven exposure and vulnerability components (RQI), and by explicitly accounting for their temporal evolution through the mRQI. One of its main strengths lies in its capacity to integrate multi-source datasets with different spatial resolutions and thematic characteristics into a*

*coherent, dimensionless index, enabling both spatial comparison across administrative units and temporal comparison within the same unit. This feature is particularly relevant for regional-scale analyses, where data heterogeneity and scale mismatches often hinder consistent risk assessment.*

*Despite these strengths, the framework also presents limitations that need to be explicitly acknowledged. At the regional scale considered, residential buildings are treated as homogeneous units in terms of construction materials and dwelling unit value, reflecting the level of data aggregation and the lack of spatially detailed building information. This simplification enables consistent comparison across municipalities but may reduce the accuracy of risk estimates at the local scale, particularly in areas with strong intra-urban heterogeneity.*

*In addition, the exclusion of susceptibility level L5 from the main analysis represents a conservative methodological choice aimed at maximizing regional representativeness, but it may lead to an underrepresentation of extreme-risk conditions at very localized scales.*

*Its performance strongly depends on the availability and quality of spatially explicit socioeconomic data, as well as on the existence of reliable landslide inventories. Consequently, the framework is most robust in contexts characterized by relatively formalized urban development, established land-use planning instruments, and consistent data collection practices, such as those typically found in European regions. In areas dominated by informal or spontaneous settlements, where building standards are weakly enforced and socioeconomic indicators are poorly documented, some components of the RQI may require alternative proxies or simplified representations. In such cases, the direct transferability of the framework without contextual adaptation may be limited.*

*From a broader research perspective, the framework should therefore be understood as a flexible methodological structure rather than a universally applicable model. Its conceptual design allows for adaptation to different territorial and socioeconomic contexts, provided that the indicators used to represent exposure and vulnerability are appropriately redefined. This makes the framework potentially relevant for applications in data-scarce or developing regions, but only after careful consideration of local conditions and data constraints. By explicitly defining its domain of applicability, this study contributes to ongoing efforts in landslide risk research to move beyond static, single-scale assessments toward more transparent, comparable, and temporally dynamic risk representations."*

*In addition, we have revised the Conclusions section to better highlight the methodological contribution of the study beyond the specific case study.*

**"7. Conclusions**

*The proposed framework demonstrates its suitability for large, spatially heterogeneous regions by addressing two critical challenges: territorial diversity and fragmented urbanization. As highlighted by Gielen et al. (2018), the Valencian Community has experienced significant urban sprawl, resulting in scattered residential areas often located in zones with contrasting hazard levels. By incorporating high-resolution cadastral data and defining dimensionless indices (RI, RSI, RQI), together with temporal trend indicators (mRQI, mRSI), the framework enables a consistent evaluation of both the magnitude and the quality of landslide risk across municipalities, overcoming the limitations of traditional approaches based on regional averages and assumptions of territorial homogeneity. In particular, the mRQI provides a meaningful indicator of the temporal evolution of risk quality at the local scale, highlighting potential progressive occupation of unsuitable land for residential housing and supporting reflection on municipal land-use practices.*

*Beyond the specific case study, the main contribution of this work lies in its methodological structure. The use of normalized, dimensionless indices combined with temporal trend analysis allows landslide risk to be decomposed into interpretable components and compared across administrative units and time periods. This approach addresses key challenges in regional-scale risk assessment, such as data*

*heterogeneity and scale inconsistencies, while remaining conceptually transparent and computationally straightforward, thereby facilitating its transfer to other regions with comparable baseline information.*

*Despite these strengths, the framework is most robust in contexts where reliable landslide inventories, cadastral information, and socioeconomic datasets are available, as is typically the case in regions characterized by formal planning systems. In areas dominated by informal settlements or limited data availability, some components of the framework may require adaptation through alternative proxies or simplified indicators. Future research should explore such adaptations, as well as the integration of population-based exposure metrics, in order to further enhance the framework's applicability across diverse geographical and socioeconomic settings.*

*Forecast-based and scenario-oriented assessments of natural hazards are essential for mitigating their impacts and supporting risk-informed territorial management. While such analyses are increasingly considered by public administrations, they are often implemented only partially or in isolation. Continued refinement of integrated and comparable risk assessment frameworks, such as the one proposed here, is therefore crucial to support more effective planning strategies and to strengthen societal capacity to anticipate and mitigate the consequences of natural hazards."*

Additionally, you may want to consider simplifying the mathematical presentation. The methodology is relatively straightforward (which is a positive feature for applicability in other contexts), but the manuscript currently presents several equations that are somewhat repetitive. A more concise set of key equations, with derivations or intermediate steps moved to an appendix/supplementary material, could improve readability.

*We thank the reviewer for this helpful suggestion. In the revised manuscript, we have simplified the mathematical presentation by retaining only the key operational equations in the main text, which are sufficient to understand and apply the proposed framework. Repetitive derivations and intermediate algebraic steps have been moved to a new Mathematical Appendix. This appendix explicitly shows how the global Risk Index (RI) can be decomposed into the Risk Surface Index (RSI) and the Risk Quality Index (RQI), and clarifies the assumptions under which further simplifications are possible. This reorganization improves readability while preserving full methodological transparency and reproducibility.*

*The text of the manuscript between lines 218 and 247 now reads as follows:*

*"...where RV is obtained through the hazard, exposure, and vulnerability of each affected dwelling unit, and exposure is defined as the constructed residential area in each risk zone ($GFA_R$) multiplied by the reconstruction cost per unit of surface (DV) (see Eq. (6)). An important feature of this index is the possibility of deriving two highly useful partial indices (see Appendix).*

*Thus, RI can be expressed as the product of two components: the Risk Surface Index (RSI) and the Risk Quality Index (RQI):*

$$RI = RSI \times RQI \tag{13}$$

$$RSI = \frac{GFA_R}{GFA} \tag{14}$$

$$RQI = \frac{RV}{H_H \times Vm \times GFA_R \times DV} \tag{15}$$

*The RSI reflects the proportion of built surface under risk relative to the total constructed area. It can approach unity in small LEs with little residential surface almost entirely exposed to landslide*

*susceptibility. The RQI, on the other hand, indicates whether the risk value approaches its theoretical maximum. Eq. (15) can be further developed and simplified assuming that residential building typologies are similar within a LE, such that vulnerability is constant ($V_i = Vm$) and DV is uniform (see Appendix).*

*Then, using the hazard probability ratio pR4 between susceptibility levels H(L3) and H(L4) defined in the LSM, and considering thar no built-up surface exists in L5, equation (15) becomes:*

$$RQI = = \frac{pR4 \times GFA_R L3 + GFA_R L4}{GFA_R} \qquad (16)$$

*This simplification of RQI clearly shows that its value depends mainly on the built surface located in high-susceptibility zones ($GFA_R L4$), since pR4 is less than one. Moreover, the RQI value provides insight into whether construction is concentrated in high-susceptibility zones (level L4) and its evolution.*

**2. Minor comments**

- **Line 50:** I am not sure whether "permanently" is the most adequate word here, as it is not possible to ensure such permanence. "This involves a complex process aimed at predicting, reducing, and permanently controlling the factors that trigger such hazards, …"

*We agree with the reviewer that the term "permanently" was too strong. The wording has been revised to avoid implying full or irreversible control of natural hazard processes.*

*"This involves a complex process aimed at predicting, reducing, and controlling the factors that trigger such hazards, …"*

- **Lines 63–64:** "Leading to landslides" seems very generic, in the sense of "urban growth leads to landslides." Maybe explain more clearly whether urban growth leads to increased landslide *hazard*, *exposure*, or *risk*. "Weak or absent administrative controls have allowed development in previously overlooked areas, leading to landslides."

*That is correct, because the problem is not really the physical landslide process, but rather exposure to it. Therefore, we modify this sentence to the following:*

*"Weak or absent administrative controls have allowed development in previously overlooked areas, leading to increased exposure to landslide hazard."*

- **Equation 2 (and others associated, such as Equation 9):** If it refers to the sum of (H × E × V) for each dwelling unit (DU), then these (H × E × V) terms should be within par entheses in the equation, correct?

*We thank the reviewer for pointing this out. We agree that the original notation could be ambiguous. The equation has been revised to explicitly indicate that the summation applies to the product of Hi, Ei and Vi for each unit i or Dwelling Unit, by including parentheses. Then, the Eq 2 will be:*

$RV_{LE} = \sum (H_{Dui} \times E_{Dui} \times V_{DUi})$

*The same clarification has been applied to related equations (e.g., Eq. 9) for consistency.*

- **Lines 176–177:** From my understanding, building resistance was assigned only based on building height, correct? Please explain this limitation. Also, I missed information on how the landslide magnitude was obtained.

*We thank the reviewer for this comment. Yes, building resistance was assigned based solely on building height, as this is the only attribute consistently available at the dwelling unit level in the cadastral database. The estimation of landslide magnitude is already described in Section 4.1.3, where LM is defined based on published damage classifications and geomorphological characteristics of the study area. To improve clarity, we have added a brief sentence (Line 446) explicitly stating that a fixed LM value was applied due to the predominance of shallow landslides. In addition, we now explicitly clarify in the methodology that building resistance is derived solely from building height, which represents a data-driven simplification and a known limitation of the approach.*

*"This fixed LM value was applied consistently across the study area due to the predominance of shallow, small-magnitude landslides".*

- **Lines 203–206:** I understand that assuming risk only at L5 would result in excessively high values. But doesn't it introduce inaccuracy to disregard buildings constructed in susceptibility level 5? I haven't taken a look at the map, so there is no way of knowing the representativeness of this class, but I imagine this is an important limitation. Maybe address this choice a bit more, and justify selecting only L4 (and not, for instance, L4+L5 combined).

*We thank the reviewer for raising this point. Indeed, restricting the analysis exclusively to susceptibility level L5 would result in very high-risk values but would also significantly reduce the representativeness of the results at the regional scale, as buildings located in L5 areas are spatially limited and constitute a very small fraction of the total housing stock. Focusing solely on this class would therefore provide a highly localized perspective rather than a comprehensive assessment of landslide risk.*

*The selection of susceptibility level L4 was motivated by the need to capture the broader range of areas where urban development intersects with potentially unstable slopes, including zones where landslide occurrence is plausible but not systematic. This choice reflects a conservative risk assessment approach, aimed at supporting land-use planning by identifying areas where precautionary measures may be warranted, even at the expense of including some overestimation. Combining L4 and L5 was considered; however, given the limited spatial extent of L5 and its strong overlap with already constrained or non-developable areas, the marginal contribution of L5 to regional-scale risk patterns was found to be low. We have now clarified this methodological choice and its implications as a limitation in the 6.2 Discussion subsection, third paragraph.*

- Some limitations could be acknowledged. For instance, the implications of assuming homogeneous buildings for all residential structures (both in terms of Dwelling Unit Value (DUV) and construction materials) and, especially, the assumption that land-use regulations will be properly followed (which may not hold in some urban contexts, particularly where enforcement capacity is limited or informality is widespread).

*We agree with the reviewer that these assumptions represent important limitations of the proposed framework. In consequence, the Discussion section (second paragraph of the 6.2 Discussion subsection) has been revised to explicitly acknowledge the assumptions regarding homogeneous building characteristics and compliance with land-use regulations, and to clarify their implications for the accuracy and transferability of the framework.*

**MATHEMATICAL APPENDIX**

An important feature of the RI index (defined in Eq. (8)) is the possibility of deriving two highly useful partial indices. By multiplying and dividing the expressions shown in Eqs. (9) and (10) by the constructed surface in risk zones ($GFA_R$), we obtain:

$$RI = \frac{GFA_R \times RV}{GFA_R \times \left(H_H \times V_m \times GFA \times DV\right)} \tag{A.1}$$

Rearranging yields:

$$RI = \frac{GFA_R}{GFA} \times \frac{RV}{H_H \times V_m \times GFA_R \times DV} \tag{A.2}$$

Thus, *RI* can be expressed as the product of two components: the *Risk Surface Index (RSI)* and the *Risk Quality Index (RQI)*:

$$RI = RSI \times RQI \tag{A.3}$$

$$RSI = \frac{GFA_R}{GFA} \tag{A.4}$$

$$RQI = \frac{RV}{H_H \times Vm \times GFA_R \times DV} \tag{A.5}$$

On the other hand, Eq. (A.5) can be further developed assuming that residential building typologies are similar within a LE, such that vulnerability is constant ($V_i = Vm$) and *DV* is uniform. Considering the hazard probability ratios (*pR*) between susceptibility levels defined in the LSM:

$$pR4 = \frac{H(L3)}{H(L4)} \tag{A.6}$$

$$pR5 = \frac{H(L4)}{H(L5)} \tag{A.7}$$

Then, for levels L3, L4, and L5:

$$RQI = \frac{H(L3) \times GFA_R L3 + H(L4) \times GFA_R L4 + H(L5) \times GFA_R L5}{GFA_R H(L4)} =$$

$$= \frac{pR4 \times GFA_R L3 + GFA_R L4 + \dfrac{1}{pR5} \times GFA_R L5}{GFA_R} = \tag{A.8}$$

$$= \frac{pR4 \times pR5 \times GFA_R L3 + pR5 \times GFA_R L4 + GFA_R L5}{GFA_R \times pR5}$$

This can be simplified when no built surface exists in L5:

$$RQI = = \frac{pR4 \times GFA_R L3 + GFA_R L4}{GFA_R} \tag{A.9}$$

This simplification of *RQI* clearly shows that its value depends mainly on the built surface located in high-susceptibility zones ($GFA_RL4$), since *pR4* is less than one. If the total *GFA* increases in the same proportion as the surface at level L4, the *RQI* value remains constant.